# Impact of System and Diagnostic Errors on Medical Litigation Outcomes: Machine Learning-Based Prediction Models

**DOI:** 10.3390/healthcare10050892

**Published:** 2022-05-12

**Authors:** Norio Yamamoto, Shintaro Sukegawa, Takashi Watari

**Affiliations:** 1Department of Epidemiology, Graduate School of Medicine, Dentistry and Pharmaceutical Sciences, Okayama University, Okayama 700-8558, Japan; 2Department of Orthopedic Surgery, Miyamoto Orthopedic Hospital, Okayama 773-8236, Japan; 3Systematic Review Workshop Peer Support Group (SRWS-PSG), Osaka 541-0043, Japan; 4Department of Oral and Maxillofacial Surgery, Kagawa Prefectural Central Hospital, Takamatsu 760-8557, Japan; gouwan19@gmail.com; 5General Medicine Center, Shimane University Hospital, Izumo 693-8501, Japan; t.watari@med.shimane-u.ac.jp; 6Division of Hospital Medicine, University of Michigan Health System, Ann Arbor, MI 48105, USA; 7Medicine Service, VA Ann Arbor Healthcare System, Ann Arbor, MI 48105, USA

**Keywords:** medical malpractice claims, litigation, diagnostic error, medical error, system error, machine learning, prediction model

## Abstract

No prediction models using use conventional logistic models and machine learning exist for medical litigation outcomes involving medical doctors. Using a logistic model and three machine learning models, such as decision tree, random forest, and light-gradient boosting machine (LightGBM), we evaluated the prediction ability for litigation outcomes among medical litigation in Japan. The prediction model with LightGBM had a good predictive ability, with an area under the curve of 0.894 (95% CI; 0.893–0.895) in all patients’ data. When evaluating the feature importance using the SHApley Additive exPlanation (SHAP) value, the system error was the most significant predictive factor in all clinical settings for medical doctors’ loss in lawsuits. The other predictive factors were diagnostic error in outpatient settings, facility size in inpatients, and procedures or surgery settings. Our prediction model is useful for estimating medical litigation outcomes.

## 1. Introduction

Medical litigation claims and costs resulting from medical errors and malpractice have increased over the past decade and have a negative impact on the health economics of both patients and medical staff [1,2,3]. Given the negative impacts of litigation on healthcare, the risk of medical litigation must be minimized for medical staff, litigation associates, and patient safety. It is better for medical staff to understand the factors that influence litigation outcomes [3,4]. System and diagnostic errors have been reported as contributing factors to malpractice claims [5,6,7,8] and recently recognized as essential issues in medical economics, health care quality, and patient safety [9,10]. In addition, previous studies have indicated the following factors associated with litigation outcomes: night shift, unnecessary surgery, sequelae, and death [5,8,11,12].

Therefore, a reliable prediction model for litigation outcomes in medical litigation research is critical for improving hospital management, which can effectively reduce plaintiff victory (medical doctor loss) [12]. However, such studies using conventional logistic regression models [13] or machine learning [14,15] to predict litigation outcomes involving medical doctors at the individual and system levels are limited. Systematic reviews have reported that prediction validity using machine learning is similar or slightly better than that of logistic regression models [16,17]. Additionally, different clinical settings, such as outpatient, inpatient, procedure, and surgery, should be investigated in each category because of their high degree of heterogeneity.

In this study, we aimed to develop and evaluate a high-prediction model for litigation outcomes in medical litigation in Japan using machine learning. Additionally, we clarified the impact of predictive factors on plaintiff victory (medical doctor losses) among comprehensive predictive factors in different clinical settings. Our hypothesis is that system and diagnostic errors contribute to medical litigation outcomes in which the doctor loses. This study will help to prepare for medical litigation, recognize modifiable factors, and improve the medical management system. 

## 2. Materials and Methods

### 2.1. Study Design and Setting

This was a retrospective cohort study based on the medical malpractice litigation records against medical doctors in Japan. We partially followed the guidelines of the transparent reporting of a multivariable prediction model for the individual prognosis or diagnosis (TRIPOD) statement [18] (Appendix A). The requirement for ethical approval was waived because the data were anonymous and obtained from a publicly available database.

### 2.2. Data Source and Study Population

We extracted data on malpractice claims against medical doctors between January 1961 and June 2017 on 29 June 2017. We used the most extensive public database in Japan (Westlaw Japan Ltd.), which includes detailed clinical information such as full text and accurate precedents. Medical claims, medical litigation, medical malpractice, diagnostic errors, wrong diagnosis, missed diagnosis, and delayed diagnosis were among the preselected keyword combinations [19]. After developing and thoroughly considering the rules in advance, five reviewers, including a lawyer and an internal medicine physician familiar with medical malpractice, performed data extraction. Of all claims extracted based on the keywords, we extracted 3430 malpractice claims. After removing duplicates (*n* = 751), applying the exclusion criteria (intentional crimes, robbery, financial difficulties, and veterinary claims; *n* = 707), and rejecting unfair lawsuits and claims against all other practitioners (*n* = 170), we extracted 1802 medical malpractice claims against medical doctors in Japan. We excluded claims with missing data as follows: patient age had 339 missing cases; clinical outcome had 55 missing cases; specialized field had 35 missing cases; facility size had 22 missing cases; time zone had 3 missing cases; and place had 2 missing cases. The final analysis included 1399 malpractice claims (Figure 1).

### 2.3. Outcomes

The primary outcome was final judgment litigation (acceptance or rejection). Acceptance meant that the medical doctor lost the medical malpractice lawsuit, whereas rejection meant that the medical doctor won it.

The secondary outcomes were clinical outcomes, including full recovery, sequelae with a permanent disorder, and death. All payments as compensation for malpractice claims were converted from the Japanese yen to US dollars using the Japanese consumer price index (115 yen to the US dollar; 12 January 2022).

### 2.4. Variables and Definitions

We selected variables based on clinical judgment and previously published literature [5,20]. We collected the following data: patient sex and age (plaintiff); the medical malpractice situation, such as the time of day (day or night shift); place where the malpractice occurred (outpatient office, emergency room, ward, and operating room); specialized field; initial diagnosis; and the institution size, such as clinic, small hospital with beds less than 200, medium hospital with beds between 200 and 399, large hospital with greater than 400 beds, or a university hospital.

The following was the detailed litigation information: board-certified doctor, the subject of the litigation (individual medical doctor or a group or hospital), the reason for the litigation (procedure, management, education, and others), and the treatment written in the precedent (medication, procedure, and others).

Medical errors were divided into diagnostic and systemic errors. A diagnostic error was defined as a delayed diagnosis, missed diagnosis, or incorrect diagnosis by an individual medical doctor [21]. Based on descriptions in the case records and following the original study [7], the system errors were categorized as follows: technical and equipment failure; clustering; inadequate policies and procedures; inefficient and non-standard processes; poor teamwork or communication; patient neglect; management problems; poor coordination of care, supervision, or education problems; unavailable expert consultation; lack of training and orientation; personnel problems, such as laziness and violations; and external interference. Multiple malpractices or complications are common, and they are not mutually exclusive. An example of a system error is shown below. Poor communication includes left–right errors on the surgical side. Management problems include a lack of proper follow-up periods and the wrong follow-up interval for the disease.

### 2.5. Statistical Analysis

Categorical variables were presented as numbers and percentages, while continuous variables were presented as medians and interquartile ranges (IQR). For group comparisons, the Mann–Whitney U test was used for continuous variables, and the chi-square test or Fisher’s exact test was used for categorical variables. Data analysis was performed using Stata SE version 17.0 (StataCorp, College Station, TX, USA). Statistical significance was set at a two-tailed *p*-value < 0.05.

Machine learning has adopted a binary classification. Supervised learning was used as the machine learning method. We developed a machine learning model that predicted the possibility of acceptance or rejection using the characteristics of the above-mentioned 64 variables. All machine learning models were implemented using Python (version 3.7.12).

We divided the data into 70% training data and 30% testing data when we first built the model. Twenty percent of the test data were further used as validation data. For hyperparameter tuning, the optimal hyperparameter was found using a grid search from the scikit-learn library using test and validation data. Then, we performed training using the training data with an optimized machine learning model and measured the performance with test data in binary classification. Cross-validation of the training datasets was performed to avoid overfitting [22]. We also performed a stratified 10-fold cross-validation to avoid data bias for each fold.

In this study, a simple linear model (logistic model) and three machine learning models, including the logistic model, decision tree, random forest, and light gradient boosting machine (LightGBM), were implemented to predict the factors contributing to plaintiff victory and to analyze the impact of predictive factors on litigation outcomes.

A logistic model is a statistical model that determines the optimal linear model coefficients to describe the relationship between the logit transformation of a binary dependent variable and one or more independent variables. Logistic models are simple forecasting approaches that provide a baseline accuracy score for comparison with other machine learning models [23].

A decision tree analysis is an analytical approach that separates predictor values in stages using binary partitioning. All the values of the predictor were evaluated as potential splits, whereas the optimal split was determined using the decrease in the entropy of the information. A classification and regression tree (CART) analysis was selected for this study [24].

Random forest is an ensemble learning algorithm that integrates multiple weak learners with decision trees to improve generalization ability [25]. It is a collection of several slightly different decision trees based on the ensemble learning bagging and has the characteristic of being less prone to overfitting.

LightGBM is a machine learning algorithm that combines a decision tree model with ensemble learning, which is a process called gradient boosting [26]. Gradient boosting is a machine learning model that eliminates the drawbacks of high calculation costs.

### 2.6. Performance Metrics and Feature Importance

Six performance metrics for machine learning used in this study were accuracy, precision, recall, specificity, F1 score, and area under the curve (AUC), which was calculated from the receiver operating characteristic (ROC) curve. These metrics are related to the classifier’s ability and calculated with true positives (TP), true negatives (TN), false negatives (FN), and false positives (FP).
(1)$$\mathrm{Accuracy}=\frac{\mathrm{TP}\;+\;\mathrm{TN}}{\mathrm{TP}\;+\;\mathrm{FP}\;+\;\mathrm{TN}\;+\;\mathrm{FN}}$$
(2)$$\mathrm{Precision}=\frac{\mathrm{TP}}{\mathrm{TP}\;+\;\mathrm{FP}}$$
(3)$$\mathrm{Recall}=\frac{\mathrm{TP}}{\mathrm{TP}\;+\;\mathrm{FN}}$$
(4)$$\mathrm{Specificity}=\frac{\mathrm{TN}}{\mathrm{TN}\;+\;\mathrm{FP}}$$
(5)$$F1\;\mathrm{score}=2\times \frac{\mathrm{Precision}\;\times \;\mathrm{Recall}}{\mathrm{Precision}\;+\;\mathrm{Recall}}$$

It is difficult to correctly interpret the output results from the machine learning model. We used the SHApley Additive exPlanation (SHAP) value, which is a unified approach, to explain the results of the machine learning models. SHAP assigns attribution values to each feature in each predictive model that are consistent and locally accurate [27]. In this study, the SHAP value was used to evaluate the feature importance.

## 3. Results

We analyzed 1399 medical litigations against medical doctors in Japan. Table 1 summarizes the demographic data for all medical claims. The median age of the patients was 33 years (IQR, 9–54), and age 0 had the highest proportion, with 253 (18.1%). The 764 (51.2%) malpractice claims resulted in acceptance (the medical doctor losing the malpractice lawsuit), and the adjusted median indemnity paid was $225,756 (IQR: 54,316–482,578). The most common patient outcome was death (56.1%), and infancy accounted for 9.3% of the deaths. Procedures or surgeries were the most common reasons for litigation, with the highest acceptance (56.1%) and residual sequence (49.1%).

Table 2 shows the clinical and litigation factors for litigation outcomes, with a crude comparison between the two groups. The top five initial diagnoses involved in malpractice claims were in the following order: malignant neoplasm, neonatal disease, trauma, procedure and postoperative complications, and acute coronary syndrome, which were not significantly associated with litigation outcome.

The factors that significantly associated with accepted claims (medical doctor loss) were as follows: clinic, small hospital (<200 beds), system error, diagnostic error, litigation subject (individual medical doctor), and sequence.

### 3.1. Machine Learning

The 30-time performance metrics evaluations of three machine learning models validated using independent test datasets showed the highest scores for LightGBM, with accuracy = 0.839 (95%CI; 0.838–0.841), precision = 0.811 (95%CI; 0.808–0.813), F1 score = 0.863 (95%CI; 0.864–0.862), and AUC = 0.894 (95%CI; 0.893–0.895) (Table 3 and Appendix A). It showed the highest scores in the decision tree, with a recall of 0.924 (0.819–0.928).

Performance metrics, excluding recall, of the logistic model for the subgroup of surgical patients showed a high performance (Appendix A). The accuracy and precision in the LightGBM of the subgroup of inpatients were 0.889 (95% CI; 0.853–0.926) and 0.881 (95% CI; 0.850–0.913), respectively (Appendix A). Recall and F1 scores had the highest scores of 0.978 (95% CI; 0.9657–1.00) and 0.906 (95% CI; 0.883–0.929), respectively. Random forest showed a high performance in the performance metrics for the subgroup of outpatients (Appendix A).

For feature importance for machine learning algorithms, we evaluated the importance rank, which indicates the importance of the input feature. The top five features important in lightGBM were system error, diagnostic error, reason for litigation (diagnosis), patient age, and era, in descending order (Figure 2). The top five features important in decision trees were system error, reason for litigation (diagnosis), diagnostic error, era, and facility size, in descending order (Appendix A). The top five features important in random forest were system error, diagnostic error, reason for litigation (diagnosis), facility size, and patient age, in descending order (Appendix A). The common features were system errors, diagnostic errors, and reasons for litigation (diagnosis).

### 3.2. Indemnity Costs

We calculated the indemnity cost for the top five predictive factors using LightGBM (Table 4). If there were more than three categories in the predictive factor, the most common acceptance category was selected. In all accepted cases, system error had the highest proportion (82.9%) and the highest indemnity cost (82.5%). Patients aged 0 years had the highest median indemnity cost (median $349,625, IQR $126,867–727,673). In the subgroups, the highest median indemnity cost was diagnostic error in the outpatient group, era (1991–1999) in the inpatient group, and patient age (age 0) in the procedures or surgery group. Diagnostic error and system error accounted for the highest proportion of total indemnity in each group.

## 4. Discussion

Machine learning has demonstrated a high performance in predicting litigation outcomes in medical litigation. System error was the most significant predictive factor for medical doctors’ loss in lawsuits in all clinical settings. The second predictive factor was diagnostic error in outpatient settings, inpatient facility size, and procedures or surgery settings.

Our prediction model had a good prediction ability using LightGBM (AUC 0.894 [95%CI; 0.893–0.895]) for all patient data. Our results are comparable to those of predictive models using machine learning with other clinical data [28,29]. It is likely that this dataset was a favorable population for prediction, because other machine learning methods also produced good results. We selected various factors, such as patient factors, medical doctor factors, and hospital factors, that could be extracted from medical lawsuit records and were likely to be associated with litigation outcomes for use in the prediction model. The assumed subgroup settings were affected as a result. Therefore, various factors must be considered to develop a good prediction model for litigation outcomes.

According to the machine learning analysis results, the system error, rather than other factors, was the most predictive factor in clinical settings. Because the legal structure and environment of medical litigation vary significantly from country to country, it may be difficult to generalize them. However, we believe that at least in the largest dataset of Japanese medical litigation, pursuing systemic problems, such as working status, a lack of standard patient safety efforts, and a lack of supervision within an organization, rather than individual medical staff errors, might result in a case being lost [10]. Other predictive factors for an accepted claim (medical doctors’ loss) using LightGBM were diagnostic error, reason for litigation (diagnosis), facility size, and patient age. These results are consistent with those of previous studies on internal medicine and orthopedic surgery [5,8,12]. Various previous studies have estimated that physician diagnostic errors in the outpatient setting may range from 3–10%, and the negative impact of diagnostic errors is a significant and urgent problem that needs to be addressed [9,30]. Thus, it is reasonable to understand that medical errors (system and diagnostic errors) are related to litigation outcomes. If a judge determines that a medical error is the basis for a lawsuit, the outcome will be unfavorable to medical providers owing to emotional appeal on the plaintiff and unprofessional negative medical behavior.

Different factors in a different order were predictive factors in three different clinical settings, namely outpatient, inpatient, and procedure or surgery. No study has examined the risk factors for litigation outcomes in different clinical settings in any department. Orthopedic surgery research has reported that the significant factors for an accepted claim are unnecessary surgery, neurological deficit, and death [12,31], although the setting types were not distinguished. The factors required for medical doctors in each clinical setting and medical errors that are likely to occur are different. Therefore, it is necessary to consider each clinical setting in medical litigation research.

### 4.1. Strengths

First, this is the first study in Japan to use a prediction model to predict litigation outcomes for medical cases. Additionally, the prediction model using LightGBM demonstrated high performance. The results of this study can be referred to by medical litigation associates and medical staff when facing medical litigation, although the ideal implication of the prediction model is a free calculator available on a website that allows missing values [28]. Second, we classified the different clinical settings into three categories: outpatient, inpatient, and procedure or surgery. Our results revealed a high degree of heterogeneity in medical litigation. This prediction model can also be used retrospectively to assess the medical quality of each setting. Third, we focused on system and diagnostic errors as predictive variables. If medical providers can recognize modifiable factors through the results of this study, it will contribute to a safer medical management system and a reduction in medical lawsuits and malpractice cases, which are associated with high socioeconomic costs and burdens. Such a situation may lead to a sincere attitude of apology and open disclosure among medical professionals, rather than concealment or contention of the patient’s claim. The results of this predictive study will provide evidence for future causal inference studies on medical litigation and patient safety.

### 4.2. Limitations

First, our data contained an inherent selection bias because the information was obtained from only a single Japanese database. In Japan, most medical litigation claims are settled out of court [32]. Because our data excluded claims dismissed before trial or settled out of court, it is difficult to generalize the findings to other countries with different legal and medical systems [5]. Second, these data did not consider legal changes in the form of trials in Japan. Japan implemented a jury system, known as the “citizen judge system”, on 21 May 2009 [33]. However, this jury system applies only to criminal trials, which were few because of the exclusion of a few criminal trials against medical professionals. Further research must determine whether our findings can be applied to medical litigation in other countries. Third, the database contained information biases. The descriptions in the database are not medical descriptions but rather the perspective of the patient, which is not always medically accurate. However, this issue can be considered a non-differential (random) misclassification that occurs equally in all study groups. Fourth, there were unmeasured factors, such as personal information on medical doctors (for example, age, sex, and graduate year), because these factors were anonymized in the database. More extensive validation using valid predictive data will be necessary in the future. The fifth limitation is the generalization performance of machine learning. Machine learning performance depends on the available training data, and deviations in input from training values can result in poor machine learning model performance. In this study, the machine learning model showed excellent internal validity; however, continuous learning and rigorous external verification will be necessary in the future. Although there were some biases and limitations in this study, our results have drawn attention to the potential impact of predictive factors on medical litigation outcomes involving medical doctors.

In conclusion, we developed a high-performance prediction model using machine learning to estimate litigation outcomes in medical litigation in Japan. Our model will be useful for estimating medical litigation outcomes.

## Figures and Tables

**Figure 1 healthcare-10-00892-f001:**
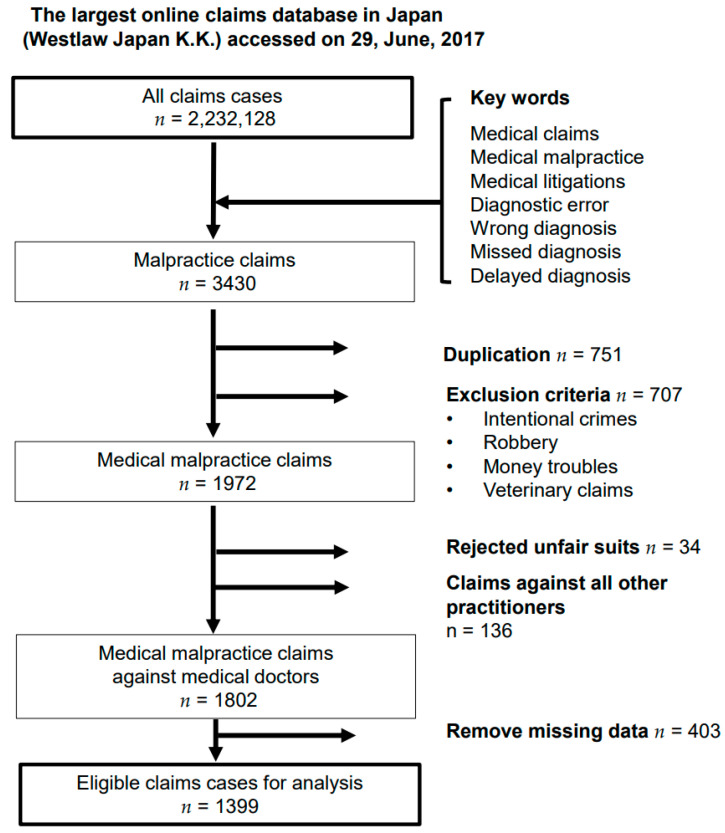
Flowchart of participant selection for the analysis.

**Figure 2 healthcare-10-00892-f002:**
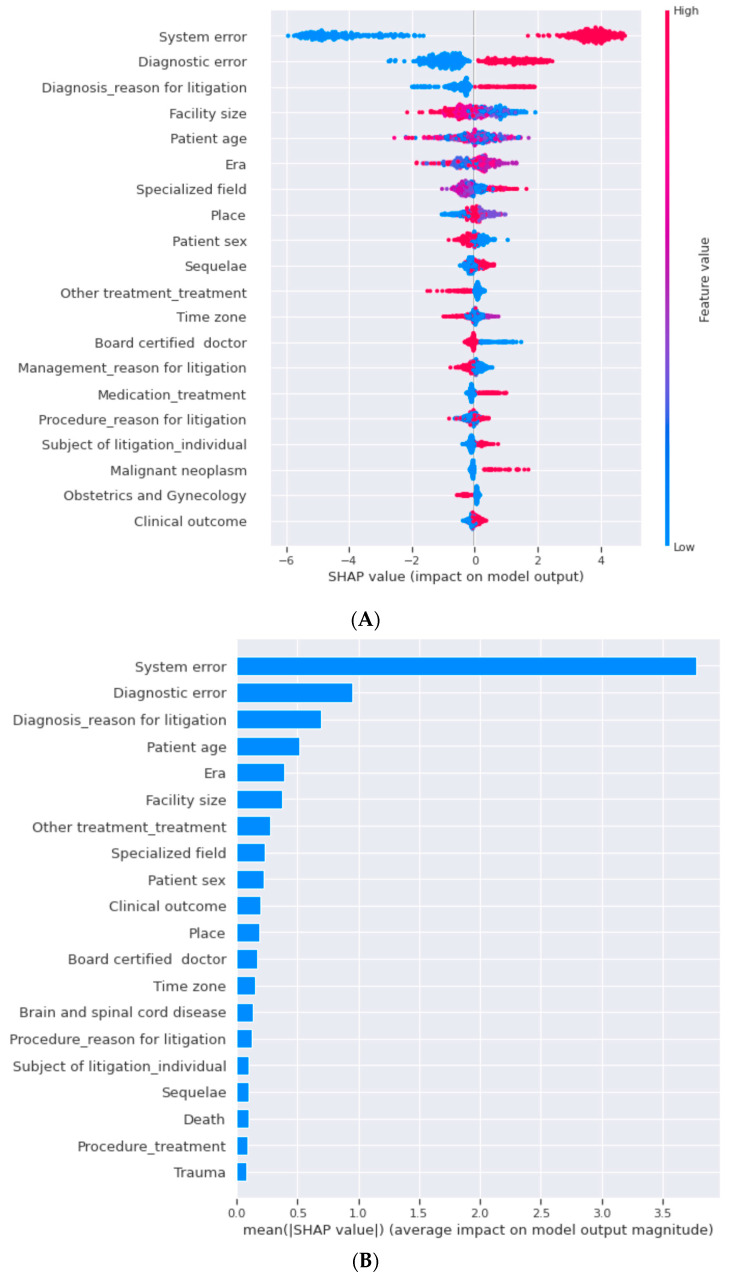
Feature importance in lightGBM with SHAP (**A**,**B**). The top five features important in lightGBM were system error, diagnostic error, the reason for litigation diagnosis, facility size, era, and patient age, in descending order.

**Table 1 healthcare-10-00892-t001:** Patient demographics and characteristics of litigation against medical doctors in Japan (*n* = 1399).

Demographics/Characteristics		Reason for Litigation		
		without Procedures or Surgery	Procedures or Surgery	
	Total	Outpatient	Inpatient	
	(*n* = 1399)	(*n* = 368)	(*n* = 424)	(*n* = 583)
Patient sex, male, *n* (%)	736 (52.6)	200 (53.4)	237 (55.9)	284 (48.7)
Patient age, median (IQR)	33 (9–54)	38 (20–54)	27 (0–55)	34 (6–52)
Adjusted total billing amount ($), median (IQR)	460,149 (202,106–799,794)	428,908 (194,291–716,004)	485,750 (241,197–777,529)	468,779 (191,367–877,166)
Subject of litigation, individual medical doctor, *n* (%)	394 (28.2)	137 (37.2)	88 (20.8)	162 (27.8)
Duration of claim (years), median (IQR)	7 (5–10)	6 (5–9)	7 (5–11)	7 (5–10)
Accepted claim, *n* (%)	764 (54.6)	196 (53.3)	231 (54.5)	327 (56.1)
Adjusted median indemnity ($), median (IQR)	236,017 (56,784–504,513)	157,069 (33,867–432,290)	265,011 (72,347–532,206)	220,008 (59,826–517,553)
Clinical outcome				
Deaths, *n* (%)	785 (56.1)	232 (63.0)	261 (61.6)	273 (46.8)
Sequelae, *n* (%)	554 (39.6)	113 (30.7)	151 (35.6)	286 (49.1)
Full recovery, *n* (%)	60 (5.3)	23 (6.3)	12 (2.8)	24 (4.1)

IQR: interquartile range. Accepted: The medical doctor has lost the case. Note: The total billing amount and median indemnity were adjusted to their 2017 equivalents using the Japanese Consumer Price Index (shown in USD, 1$ = ¥115, 12 January 2022).

**Table 2 healthcare-10-00892-t002:** Clinical and litigation factors on litigation outcomes.

		Accepted (*n* = 764)	Rejected (*n* = 635)	*p*-Value
Patient sex, male, *n* (%)		395 (51.7)	341 (53.7)	0.485
Patient age, median (IQR)		32 (11–53)	34 (7.5–56)	0.625
Initial diagnoses (Top 5 involved in malpractice claims), *n* (%)				
	Malignant neoplasm (*n* = 115)	60 (7.9)	55 (8.7)	0.625
	Neonatal disease (*n* = 110)	66 (8.6)	44 (6.9)	0.273
	Trauma (*n* = 109)	64 (8.4)	45 (7.1)	0.423
	Procedure and postoperative complications (*n* = 67)	42 (5.5)	25 (3.9)	0.208
	Acute coronary syndrome (*n* = 66)	37 (4.8)	29 (4.6)	0.899
Specialty, *n* (%)				
	Surgical specialties	439 (57.5)	335 (52.8)	0.084
	Non-surgical specialties	202 (26.4)	202 (31.8)	0.028
Place, *n* (%)				
	Outpatient office	145 (19.0)	132 (20.8)	0.419
	Emergency room	51 (6.7)	40 (6.3)	0.828
	Ward	231 (30.2)	193 (30.4)	0.953
	Operation room	327 (42.8)	256 (40.3)	0.355
Facility size, *n* (%)				
	Clinic	223 (29.2)	137 (21.6)	0.001
	Small hospital (<200 beds)	166 (21.7)	110 (17.3)	0.043
	Medium hospital (200–399 beds)	264 (34.6)	243 (38.3)	0.163
	Large (>400 beds) or university hospital	111 (14.5)	145 (22.8)	<0.001
Time, *n* (%)				
	Day time	480 (62.8)	379 (59.7)	0.247
	Night shift	121 (15.8)	83 (13.1)	0.149
Error type, *n* (%)				
	System error	634 (83.0)	127 (20.0)	<0.001
	Diagnostic error	377 (49.3)	205 (32.3)	<0.001
Subject of litigation, *n* (%)				
	Individual medical doctor	238 (31.2)	156 (24.6)	0.007
	Group or hospital	548 (71.7)	491 (77.3)	0.02
Era *		4/11/60/135/174/311/65/4	0/6/67/162/114/197/85/4	NA
Clinical outcome, *n* (%)				
	Deaths	411 (53.8)	374 (58.9)	0.058
	Sequelae	321 (42.0)	233 (36.7)	0.048
	Full recovery	32 (4.2)	28 (4.4)	0.895

Accepted: Medical doctors lost the case. Rejected: Medical doctors won the case. IQR, interquartile range; NA, not available because of a zero event. * 10-year interval, 1940–1949, 1950–, 1960–, 1970–, 1980–, 1990–, 2000–, 2010–2017.

**Table 3 healthcare-10-00892-t003:** Machine learning and logistic model-based prediction models for litigation outcomes in all cases.

	LightGBM		Decision Tree		Random Forest		Logistic Model
Accuracy	0.839	(0.838–0.841)	0.825	(0.823–0.826)	0.832	(0.831–0.834)	0.826 (0.825–0.827)
Precision	0.811	(0.808–0.813)	0.787	(0.781–0.794)	0.810	(0.808–0.813)	0.810 (0.809–0.811)
Recall	0.924	(0.819–0.928)	0.935	(0.920–0.950)	0.907	(0.901–0.913)	0.893 (0.894–0.891)
F1 score	0.863	(0.864–0.862)	0.853	(0.850–0.856)	0.855	(0.853–0.857)	0.849 (0.848–0.850)
AUC	0.894	(0.893–0.895)	0.874	(0.872–0.876)	0.894	(0.893–0.896)	0.881 (0.881–0.882)

The value is described as the predictive ability (95% CI). CI, confidence interval; AUC, area under the curve.

**Table 4 healthcare-10-00892-t004:** Impact of the top five predictive factors on medical doctor loss (accepted claims).

Factors	*n* (%)	Indemnity ($), Median (IQR)	Total Indemnity ($)	Proportion of All Total Indemnity in Each Group (%)
All Cases (*n* = 764)				
System error	634 (82.9)	212,971 (53,651–450,478)	201,959,117	82.5
Diagnostic error	377 (49.3)	248,534 (59,279–507,662)	133,875,865	54.6
Reason for litigation: diagnosis	186 (24.3)	202,639 (67,344–482,423)	60,084,309	24.5
Facility size (medium hospital)	264 (34.5)	237,931 (67,344–482,423)	86,910,186	35.5
Patient age (age 0)	134 (17.5)	349,625 (126,867–727, 673)	59,320,694	24.2
Subgroups				
Outpatient (*n* = 196)				
System error	108 (55.1)	82,920 (28,562–346,612)	26,098,691	46.6
Diagnostic error	150 (76.5)	206,364 (39,476–463,340)	49,727,644	88.9
Patient age (age 0)	10 (5.1)	95,216 (27,170–804,262)	4,101,603	7.3
Era (1991–1999)	85 (43.3)	184,002 (31,533–499,561)	28,974,661	51.8
Treatment: other treatments	55 (28.0)	59,279 (31,676–271,776)	12,369,668	22.1
Inpatient (*n* = 231)				
System error	209 (90.4)	242,469 (73,847–492,951)	70,966,535	91.0
Facility size (medium hospital)	100 (43.2)	269,137 (89,484–513,205)	32,763,778	42.0
Era (1991–1999)	89 (38.5)	300,128 (57,342–531,114)	31,572,902	40.4
Diagnostic error	109 (47.1)	297,139 (103,474–565,210)	42,819,638	54.9
Sequence	138 (59.7)	237,485 (62,423–408,069)	39,358,055	50.4
Procedures or surgery (*n* = 327)				
System error	309 (94.4)	206,791 (68,185–478,915)	102,194,174	94.4
Facility size (medium hospital)	113 (34.5)	254,086 (108,194–643,171)	25,272,862	23.3
Patient age (age 0)	60 (18.3)	481,070 (278,112–833,305)	32,018,881	29.6
Diagnostic error	112 (34.2)	249,226 (76,811–486,514)	40,354,005	37.3
Era (1991–1999)	134 (40.9)	283,772 (77,425–620,218)	52,708,845	48.7

## Data Availability

The data that support the findings of this study are available from the last author, T.W., upon reasonable request.

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
