# Peer review of "Impact of System and Diagnostic Errors on Medical Litigation Outcomes: Machine Learning-Based Prediction Models"

_healthcare, 2022, doi:10.3390/healthcare10050892_

Round 1

Reviewer 1 Report

# Paper discusses the high-performance prediction model using machine learning to estimate litigation outcomes in medical litigation in Japan. This is appreciated.

# At last of Introduction Section, please provide "Major Contributions" of this paper. Also, provide overall organization of  this paper after such contributions.

#Add "Related Work" Section after the introduction section so that proposed work can be justified with findings & observations.

# Add more references and cite below article to improve the readability of your paper:

1) Arya, K.V. and Bhadoria, R.S., 2019. The Biometric Computing: Recognition and Registration. CRC Press.

2) Singh, L. K., Garg, H., Khanna, M., & Bhadoria, R. S. (2021). An enhanced deep image model for glaucoma diagnosis using feature-based detection in retinal fundus. Medical & Biological Engineering & Computing, 59(2), 333-353.

Author Response

Response to Reviewer 1’s Comments:

Reviewer 1 

Comment R1-1:

# Paper discusses the high-performance prediction model using machine learning to estimate litigation outcomes in medical litigation in Japan. This is appreciated.

Response R1-1:

Thank you for this positive comment. 

Comment R1-2:

# At last of Introduction Section, please provide "Major Contributions" of this paper. Also, provide overall organization of  this paper after such contributions.

Response R1-2:

We have added those bits of information as follows. 

Lines 55–59

This study will help to prepare for medical litigation, recognize modifiable factors, and improve the medical management system. The remainder of this paper is organized as follows. Section 2 describes the materials and methods used in this study. Section 3 presents and analyzes the results obtained. Section 4 discusses the overall study, outlines its strengths and limitations, and presents concluding remarks.

Comment R1-3:

#Add "Related Work" Section after the introduction section so that proposed work can be justified with findings & observations.

Response R1-3:

Thank you for your important comments. Although we do not have a distinct “Related Work” section, we have actually cited related previous studies in the introduction section to justify our logic. Therefore, we believe that adding an actual section titled “Related Work” may be redundant. The related studies cited are as follows:

Artificial intelligence

  1. Sukegawa, S.; Fujimura, A.; Taguchi, A.; et al. Identification of Osteoporosis Using Ensemble Deep Learning Model with Panoramic Radiographs and Clinical Covariates [Sci. rep., 2022, 12(1):6088].
  2. Yamamoto, N.; Sukegawa, S.; Yamashita, K.; Manabe, M.; Nakano, K.; Takabatake, K.; Kawai, H.; Ozaki, T.; Kawasaki, K.; Nagatsuka, H.; et al. Effect of Patient Clinical Variables in Osteoporosis Classification Using Hip X-Rays in Deep Learning Analysis. Medicina (Kaunas). 2021, 57, 846. DOI:10.3390/medicina57080846.
  3. Yamamoto, N.; Sukegawa, S.; Kitamura, A.; Goto, R.; Noda, T.; Nakano, K.; Takabatake, K.; Kawai, H.; Nagatsuka, H.; Kawasaki, K.; et al. Deep Learning for Osteoporosis Classification Using Hip Radiographs and Patient Clinical Covariates. Biomolecules. 2020, 10, 1534. DOI:10.3390/biom10111534.
  4. Sukegawa, S.; Yoshii, K.; Hara, T.; Yamashita, K.; Nakano, K.; Yamamoto, N.; Nagatsuka, H.; Furuki, Y. Deep Neural Networks for Dental Implant System Classification. Biomolecules. 2020, 10, 984. DOI:10.3390/biom10070984.
  5. Sukegawa, S.; Matsuyama, T.; Tanaka, F.; et al. Evaluation of Multi-Task Learning in Deep Learning-Based Positioning Classification of Mandibular Third Molars [Sci. rep., 2022, 12(1):684].

Medical litigation or medical error

  1. Watari T, Tokuda Y, Amano Y, Onigata K, Kanda H. Cognitive Bias and Diagnostic Errors among Physicians in Japan: A Self-Reflection Survey. Int J Environ Res Public Health. 2022;19(8):4645.
  2. Notomi K, Harada T, Watari T, Hiroshige J, Shimizu T. Misdiagnosis Due to False-Positive Detection of Pneumococcal Urinary Antigen. Eur J Case Rep Intern Med. 2022;9(2):003198.
  3. Enomoto K, Kosaka C, Kimura T, et al. Pharmacists can improve diagnosis and help prevent diagnostic errors [published online ahead of print, 2022 Jan 31]. Diagnosis (Berl). 2022;10.1515/dx-2021-0138. doi:10.1515/dx-2021-0138
  4. Harada T, Miyagami T, Watari T, et al. Barriers to diagnostic error reduction in Japan [Published online ahead of print, 2021 Jun 30]. Diagnosis (Berl). 2021;10.1515/dx-2021-0055. doi:10.1515/dx-2021-0055
  5. Watari T. Malpractice Claims of Internal Medicine Involving Diagnostic and System Errors in Japan. Intern Med. 2021;60(18):2919-2925. 
  6. Harada T, Miyagami T, Watari T, et al. Analysis of diagnostic error cases among Japanese residents using diagnosis error evaluation and research taxonomy. J Gen Fam Med. 2021;22(2):96-99.
  7. Otsuki K, Watari T. Characteristics and Burden of Diagnostic Error-Related Malpractice Claims in Neurosurgery. World Neurosurg. 2021;148:e35-e42.
  8. Watari T, Tokuda Y, Mitsuhashi S, et al. Factors and impact of physicians' diagnostic errors in malpractice claims in Japan. PLoS One. 2020;15(8):e0237145. 
  9. Yamamoto N, Watari T, Shibata A, Noda T, Ozaki T. The impact of system and diagnostic errors for medical litigation outcomes in orthopedic surgery [published online ahead of print, 2021 Dec 6]. J Orthop Sci. 2021;S0949-2658(21)00371-7.

Comment R1-4:

# Add more references and cite below article to improve the readability of your paper:

Response R1-4:

Thank you for recommending these studies. We have reviewed and cited them as follows.

Line 43

However, such studies using conventional logistic regression models [13] or machine learning [15,15] to predict litigation outcomes involving medical doctors at the individual and system levels are limited.

  1. Lage-Freitas A, Allende-Cid H, Santana O, Oliveira-Lage L. Predicting Brazilian Court Decisions. PeerJ Comput Sci. 2022;8:e904. Published 2022 Mar 25. doi:10.7717/peerj-cs.904
  2. Sert MF, Yıldırım E, Haşlak İ. Using Artificial Intelligence to Predict Decisions of the Turkish Constitutional Court. Social Science Computer Review, 2021, doi:10.1177/08944393211010398

Reviewer 2 Report

The manuscript is well-written and the general structure of the study well designed. One of the main limitations of the study is related to the fact that the phenomenon of medical malpractice is closely dependent on the context in which it is examined. Even with these limitations, the work carried out could be of some relevance to an international audience. For this reason there are some points that should be clarified. 

-lines 34-35: the authors repeat the term medical in the same sentence 5 times; in order to make the reading easier it would be advisable to use synonyms or avoid such repetitions;

-line 61: the authors write" ...we partially followed the guidelines ...",it would be advisable to better explain in which parts they have followed the above-mentioned guidelines and in which parts they have diverged from them, clarifying the underlying reasons; 

-line 70: the authors state : "... were among the preselected keyword combination ...", there are other keyword combination not cited? if there please explain; 

-lines 107-113: in order to identify the different types of system errors, it would be useful to give examples of each or a brief description of them.

As a general suggestion, authors should propose a description of what they believe could be the positive practical implications of the proposed predictive model and its possible advantages. 

Author Response

Response to Reviewer 2’s Comments:

Reviewer 2

Comment R2-1:

The manuscript is well-written and the general structure of the study well designed. One of the main limitations of the study is related to the fact that the phenomenon of medical malpractice is closely dependent on the context in which it is examined. Even with these limitations, the work carried out could be of some relevance to an international audience. For this reason there are some points that should be clarified. 

Response R2-1:

Thank you for this positive comment. 

Comment R2-2:

-lines 34-35: the authors repeat the term medical in the same sentence 5 times; in order to make the reading easier it would be advisable to use synonyms or avoid such repetitions;

Response R2-2:

Thank you for this advice. We have revised the sentence accordingly.

Line 34

Given the negative impacts of litigation on healthcare, the risk of medical litigation must be minimized for medical staff, litigation associates, and patient safety.

Comment R2-3:

-line 61:the authors write" ...we partially followed the guidelines...",it would be advisable to better explain in which parts they have followed the above-mentioned guidelines and in which parts they have diverged from them, clarifying the underlying reasons; 

Response R2-3:

Thank you for this important suggestion. To provide more clarity, we have added Table S1, which presents the TRIPOD checklist.

Line 63

We partially followed the guidelines of the transparent reporting of a multivariable prediction model for individual prognosis or diagnosis (TRIPOD) statement [18] (Table S1).

Line 330

Table S1: TRIPOD checklist: Prediction model development

Comment R2-4:

-line 70: the authors state : "... were among the preselected keyword combination ...", there are other keyword combination not cited? if there please explain; 

Response R2-4:

There are no the other keyword combinations.

Comment R2-5:

-lines 107-113: in order to identify the different types of system errors, it would be useful to give examples of each or a brief description of them.

Response R2-5:

Thank you for this suggestion. We have added the following sentence accordingly.

Line 116

An example of a system error is shown below. Poor communication includes left-right errors on the surgical side. Management problems include lack of proper follow-up periods and the wrong follow-up interval for the disease.

Comment R2-6:

As a general suggestion, authors should propose a description of what they believe could be the positive practical implications of the proposed predictive model and its possible advantages. 

Response R2-6:

We agree with your suggestion, and have added the following information accordingly.

Line 288

The results of this study can be referred to by medical litigation associates and medical staff when facing medical litigation, although the ideal implication of the prediction model is a free calculator available on a website that allows missing values [28].

Line 295

If medical providers can recognize modifiable factors through the results of this study, it will contribute to a safer medical management system and a reduction in medical lawsuits and malpractice cases, which are associated with high socioeconomic costs and burden. Such a situation may lead to a sincere attitude of apology and open disclosure among medical professionals rather than concealment or contention of the patient’s claim.